# Prevalence, Mechanism, Genetic Diversity, and Cross-Resistance Patterns of Methicillin-Resistant *Staphylococcus* Isolated from Companion Animal Clinical Samples Submitted to a Veterinary Diagnostic Laboratory in the Midwestern United States

**DOI:** 10.3390/antibiotics11050609

**Published:** 2022-04-30

**Authors:** Mehmet Cemal Adiguzel, Kayla Schaefer, Trevor Rodriguez, Jessica Ortiz, Orhan Sahin

**Affiliations:** 1Department of Veterinary Diagnostic and Production Animal Medicine, College of Veterinary Medicine, Iowa State University, Ames, IA 50011, USA; mcemal.adiguzel@atauni.edu.tr (M.C.A.); kaylaschaeferdvm@gmail.com (K.S.); trevor.rodriguez@westernu.edu (T.R.); jessicaortiz92.jo@gmail.com (J.O.); 2Department of Microbiology, Faculty of Veterinary Medicine, Ataturk University, Erzurum 25240, Turkey

**Keywords:** methicillin-resistant *Staphylococcus*, companion animals, clinical cases, antimicrobial resistance, genetic diversity, PBP2a, *mec* genes, veterinary diagnostic laboratory

## Abstract

Methicillin-resistant *Staphylococcus* (MRS) is a leading cause of skin and soft tissue infections in companion animals, with limited treatment options available due to the frequent cross-resistance of MRS to other antibiotics. In this study, we report the prevalence, species distribution, genetic diversity, resistance mechanism and cross-resistance patterns of MRS isolated from companion animal (mostly dog and cat) clinical cases submitted to Iowa State University Veterinary Diagnostic Laboratory (ISU VDL) between 2012 and 2019. The majority of isolates were identified as *Staphylococcus pseudintermedius* (68.3%; 2379/3482) and coagulase-negative *Staphylococcus* (CoNS) (24.6%; 857/3482), of which 23.9% and 40.5% were phenotypically resistant to methicillin, respectively. Cross resistance to other β-lactams (and to a lesser extent to non-β-lactams) was common in both methicillin-resistant *S. pseudintermedius* (MRSP) and CoNS (MRCoNS), especially when oxacillin MIC was ≥4 μg/mL (vs. ≥0.5–<4 μg/mL). The PBP2a protein was detected by agglutination in 94.6% (521/551) MRSP and 64.3% (146/227) MRCoNS. A further analysis of 31 PBP2a-negative MRS isolates (all but one MRCoNS) indicated that 11 were *mecA* gene-positive while 20 were negative for *mecA* and other *mec* genes by PCR. The resistance to last-resort anti-staphylococcal human drugs (e.g., tigecycline, linezolid, vancomycin) among the MRS tested was none to very low. Even though genotyping indicated an overall high level of genetic diversity (87 unique PFGE patterns and 20 MLST types) among a subset of MRSP isolates tested (*n* = 106), certain genotypes were detected from epidemiologically connected cases at the same or different time points, suggesting persistence and/or nosocomial transmission. These results indicate a relatively high prevalence of MRS from companion animals in the Midwestern US; therefore, it is important to perform routine susceptibility testing of *Staphylococcus* in veterinary clinical settings for the selection of appropriate antimicrobial therapy.

## 1. Introduction

Staphylococci are among the leading causes of skin, ear, and wound infections in dogs, cats, and other companion animals, and methicillin-resistant *Staphylococcus* (MRS) is of high concern because of the difficulty in treating associated infections [1,2,3]. *Staphylococcus pseudintermedius* is a coagulase-positive bacterium, which is primarily a commensal organism on the skin and mucocutaneous sites of dogs, but it has also been found in the skin microflora of other companion animals, and is the primary etiologic agent of opportunistic staphylococci infections in canine hosts [1,3,4,5]. In recent years, MRS—which includes methicillin-resistant *S. pseudintermedius* (MRSP), methicillin-resistant coagulase-negative *Staphylococcus* (MRCoNS) such as *Staphylococcus epidermidis* (MRSE), and other coagulase-positive *Staphylococcus* (MRCoPS) such as methicillin-resistant *Staphylococcus aureus* (MRSA)—has been globally on the rise [1,2,3,6,7,8,9,10,11].

While *S. aureus* is considered the predominant pathogenic species in humans, *S. pseudintermedius* has also been detected in human clinical submissions with an increasing frequency [12,13,14,15]. This increased reporting has most likely occurred due to the improvement in diagnostic microbiological methods as many *S. pseudintermedius* infections were previously misidentified as *S. aureus* [12,13,15]. There is evidence for the zoonotic transmission of *S. pseudintermedius* and other staphylococci between companion animals and humans, with *S. pseudintermedius* being a potentially important emerging zoonosis because the organism commonly colonizes dogs and shares similar virulence factors with *S. aureus* [3,15,16,17,18,19].

Methicillin resistance in staphylococci is typically mediated by the *mecA* gene that encodes penicillin-binding protein 2a (PBP2a), a protein with a low affinity for most β-lactam antibiotics [20,21,22,23]. The *mecA* gene is carried on a mobile genetic element, the staphylococcal cassette chromosome *mec* (SCC*mec*) [21,22,23]. Recently, other variants of the *mec* gene have been described, including *mecB*, *mecC*, and *mecD* [21,24,25,26,27,28,29,30]. The detection of *mecA* by PCR is the gold standard for methicillin-resistance identification, but use of latex agglutination testing (LAT) and immunochromatographic assays directed toward PBP2a provides more rapid methods of detection with a high sensitivity and specificity [20,21,27,31,32,33]. MRS is frequently resistant to many other antibiotics besides β-lactams, which poses a challenge for treatment [3,6,7,11,34,35,36]. Multidrug resistance is especially prominent in MRSP and certain MRSP clones and is often seen with resistance to other antibiotic classes important to veterinary medicine, including fluoroquinolones, tetracyclines, aminoglycosides, macrolides, lincosamides, chloramphenicol, trimethoprim, and sulfonamides [5,9,13,16,18,36,37,38,39,40,41,42]. The increased prevalence of MRS and cross-resistance to commonly used antibiotics has been linked to antibiotic treatment practices in dogs [2,5,18]. There are also limited antibiotic susceptibility testing data available for antibiotics used in human medicine for *Staphylococcus* species, such as *S. pseudintermedius*, which are predominant in animal hosts [13].

Genotyping is a commonly utilized tool for the detailed investigation of MRS isolates from companion animals (e.g., clinical isolates from dogs) for various epidemiological and population genetic purposes around the world [3,5,9,11,12,17,34,40,41,43,44,45,46,47,48,49,50]. Among the molecular subtyping methods, pulsed-field gel electrophoresis (PFGE), multilocus sequence typing (MLST), SCC*mec* typing, staphylococcal protein A (*spa*) typing, and more recently whole-genome sequencing (WGS) are the most commonly used. Such studies indicated that even though MRS, especially MRSP, shows a high phylogenic and genetic diversity, overall, it has a clonal population structure with the predominance of several epidemic clones in certain geographic regions and across the globe. For example, MLST sequence types (STs) of MRSP ST71, and to a lesser extent ST45 and ST258, are the most common clones in Europe and many other parts of the world, while ST68 is the main lineage in North America. Findings from these studies also show a much higher genetic heterogeneity among methicillin-susceptible *S. pseudintermedius* and other *Staphylococcus* spp. compared with their methicillin-resistant populations. In addition, several studies have also indicated an association between certain phenotypic characteristics (e.g., antimicrobial resistance and virulence) and distinct genotypes [17,34,35,44,45,47,51,52,53]. For example, there were significant differences among the major MLST clonal complexes (e.g., CC45, CC71, and CC258) of MRSP isolates in their resistance profiles to some important antibiotics, including enrofloxacin, gentamicin, trimethoprim/sulfamethoxazole, and tetracycline [45]. Another study [44] also showed a close correlation between genotypes (based on MLST, SCC*mec*, and *spa*) typing) and phenotypes (i.e., antimicrobial resistance profiles for several antibiotics) among MRSP isolates from healthy dogs. Likewise, the types of SCC*mec* carried by different MRSP clonal lineages were found to vary significantly as reported in other studies around the world [5,7,45,50]. The close association between genotypes and antimicrobial resistance phenotypes may be an important consideration for devising a successful MRS mitigation program in the end. Similarly, some studies found an association between genotypes (e.g., MLST sequence types, SCC*mec* types, PFGE types) and certain virulence-associated traits, such as biofilm and slime production, toxins, and other virulence gene carriages or their expression levels, as well as accessory gene regulatory (*agr*) groups in MRSP isolates from companion animals [17,51,52].

Continued surveillance in veterinary settings is important for both gauging the extent of overall resistance threat posed by MRS and for detecting emergence of new MRS clones of high concern (e.g., cross-resistance to last-resort antibiotics) at local and global levels. The goal of this study was to determine the prevalence, species distribution, genetic diversity, methicillin resistance mechanisms, and cross-resistance patterns of MRS isolates recovered from companion animal clinical cases submitted to Iowa State University Veterinary Diagnostic Laboratory (ISU VDL) from 2012 to 2019.

## 2. Results and Discussion

### 2.1. Prevalence of MRS

Of a total of 3482 *Staphylococcus* isolates recovered from companion animal clinical submissions at ISU VDL between 2012 and 2019, 68.3% were *S. pseudintermedius*, 17.2% were CoNS (excluding *S. epidermidis*), 7.1% were *S. epidermidis*, 5.8% were *S. aureus*, and 1.6% were CoPS (excluding *S. aureus* and *S. pseudintermedius*). The prevalence of methicillin resistance in *Staphylococcus* spp. was 27.9% overall, 23.9% in *S. pseudintermedius*, 40.2% in CoNS, 42.7% in *S. epidermidis*, 22.9% in *S. aureus*, and 16.4% in CoPS (Figure 1), indicating a moderate to high level of resistance. As can be seen in Figure 2, the prevalence of MRS by year of isolation showed an overall upward trend, being the lowest in 2013 (20%) and highest in 2018 (34.5%).

Overall, comparable rates of species distribution (i.e., predominantly *S. pseudintermedius* followed by CoNS and less frequently by other species) and the methicillin resistance of *Staphylococcus* spp. from companion animal (largely dogs) clinical specimens have been reported by other studies from different geographic regions of the world during the last two decades. The frequencies of methicillin resistance ranged from 0% to 57% in clinical *S. pseudintermedius* isolates and were between 0.5% and 66% in CoNS, with an overall increasing trend in resistance over time [5,7,8,10,12,17,18,35,36,40,42,43,49,54,55,56,57,58]. In general, methicillin resistance rates in non-disease associated *S. pseudintermedius* and other *Staphylococcus* spp. isolates from companion animals were reported to be lower than those in isolates from infection sites [10,17,47]. The variation observed in MRS prevalence among studies could be due to influences based on patient population, geography, study periods, and the methods used for sampling. In this study, the moderate–high MRS prevalence observed could be due to selection bias because samples submitted to ISU VDL come from clinical cases that may have failed to respond to initial treatment; therefore, these isolates may be more likely to be positive for methicillin resistance.

### 2.2. Cross-Resistance Patterns of MRS

Antibiotic cross-resistance patterns of *S. pseudintermedius* and coagulase-negative staphylococci (CoNS; including *S. epidermidis*) based on oxacillin (Oxa) minimum inhibitory concentration (MIC) values are shown in Table 1 and Table 2, respectively. The cross-resistance of coagulase-positive staphylococci (CoPS; including *S. aureus*) was not analyzed due to the relatively small numbers of isolates found within these groups in this study (Figure 1). The isolates were divided into three categories for cross-resistance analysis based on Oxa MICs (susceptible and resistant by two different breakpoints). This was carried out because the oxacillin CLSI resistance breakpoint used for *S. pseudintermedius*/CoNS (MIC ≥ 0.5) was much lower than that used for *S. aureus* (MIC ≥ 4). The purpose of this was to determine if the in vitro cross-resistance patterns were correlated with the degree (i.e., MIC) of Oxa resistance. It should be emphasized that MRS isolates are automatically considered clinically resistant to all of the β-lactam antibiotics (except for imipenem) tested in this study per CLSI guidelines [59]; however, these antibiotics were still included in the analysis for the evaluation of their in vitro susceptibility levels by Oxa MIC.

As can be seen from the data presented in these tables, methicillin resistance in both *S. pseudintermedius* and CoNS was well-associated with in vitro resistance to other β-lactam antibiotics (except for imipenem), and the degree of cross-resistance was especially high when the Oxa MIC was ≥4 µg/mL. Similar but relatively less prominent cross-resistance patterns were also observed with the vast majority of non-β-lactam antibiotics, including aminoglycosides, fluoroquinolones, tetracyclines, lincosamide, macrolide, phenicol, and trimethoprim/sulfamethoxazole. Many other studies around the world also reported high levels of cross-resistance among MRS to β-lactams and other antibiotic classes, especially in MRSP from dogs [3,5,6,7,9,11,13,16,17,18,34,35,36,37,38,39,40,41]. However, our study further analyzed the degree of cross-resistance by Oxa MIC level. The increased prevalence of MRS and cross-resistance to commonly used antibiotics is of high concern for therapeutic options and has been linked to antibiotic treatment practices in dogs [2,5,18].

In human medicine, the “MRSA Expert Rule”, devised by the CLSI and the European Committee on Antimicrobial Susceptibility Testing (EUCAST), recommends that all MRSA isolates are reported as resistant to all β-lactams with a few exceptions, and has been applied to veterinary medicine for MRSP and MRCoNS without sufficient microbiological or clinical evidence [1,59,60,61]. Since MRSP and MRCoNS have a much lower clinical breakpoint for methicillin resistance (≥0.5 μg/mL) than MRSA (≥4 μg/mL), it was suggested that MRSP and MRCoNS might be overreported as resistant to other β-lactams in veterinary medicine [1]. Our analysis appears to support this view, at least for some antibiotics, including cefazolin, cefoxitin (for MRSP only), and cephalothin, since the in vitro cross-resistance to these antimicrobials was not substantially higher when Oxa MIC was ≥0.5–<4 μg/mL compared with that in Oxa-susceptible isolates; whereas, the cross-resistance was quite remarkable when the Oxa MIC was ≥4 μg/mL as compared with Oxa-susceptible isolates (Table 1 and Table 2). It should be emphasized that convincing in vivo data are needed to definitively determine whether these drugs would be efficacious for treating clinical MRS infections (caused by resistant isolates having relatively lower Oxa MIC) in animals.

### 2.3. Resistance to Anti-Gram-Positive Antimicrobials in MRS

A randomly selected subset (*n* = 236) of more recent (2016 through 2018) MRS isolates (mostly *S. pseudintermedius* and CoNS) were tested to determine if they also developed resistance to a broader selection of anti-Gram-positive antimicrobials included in the Sensititre NARMS panel (Table 3). The isolates displayed none to very low levels of resistance to some of the first-line antibiotics, including anti-MRSA and other important drugs used in human medicine, such as tigecycline, daptomycin, linezolid, vancomycin, and quinupristin/dalfopristin. Non-susceptibility was observed in only one isolate (CoNS, *Staphylococcus xylosus*) to tigecycline, five isolates (mostly CoNS) to daptomycin, and nine isolates to quinupristin/dalfopristin (Table 3). These data indicated that MRS isolates from companion animals in the current study remain largely susceptible to newer anti-MRSA and anti-Gram-positive antibiotics of importance for human medicine, and suggests that these drugs could still be effective against MRS infections of a zoonotic nature. Similarly, resistance to last-resort antibiotics of importance for human medicine in *Staphylococcus* spp. (including *S. pseudintermedius* and CoNS among other species, and both methicillin-susceptible and -resistant isolates) derived from different hosts, including companion animals and humans, was reported to be between very low and absent by other investigators around the world [7,9,13,62,63,64].

### 2.4. Mechanism of Methicillin Resistance in MRS

The overall PBP2a prevalence in MRS isolates tested in this study was 84.3%, 94.6% in *S. pseudintermedius*, 63.3% in MRCoNS, 68.6% in MRSE, 93.8% in MRSA, and 93.0% in MRCoPS. This finding indicated that *mecA*/PBP2a was responsible for the vast majority of methicillin resistance in MRCoPS (including *S. pseudintermedius*), while a significant proportion of MRCoNS isolates (including MRSE) appeared to have a different resistance mechanism (see below for possible explanations for this observation). A relatively low sensitivity of immunochromatographic assays for the detection of PBP2a in MRCoNS was also reported in a previous study, especially when the testing was performed without prolonged incubation or the cefoxitin induction of cultures [20].

A total of 36 MRS isolates (per repeated MIC results; all but one CoNS) that were initially tested as PBP2a-negative were available for further characterization. Standard (i.e., after 24 h incubation period) repeat PBP2a testing on these isolates confirmed the initial results. However, following an additional 24 h incubation of these cultures, five of them yielded a positive PBP2a reaction, of which four were also positive for *mecA* PCR and one was negative for *mecA* and the other genes tested (Table 4). The PBP2a-positive and *mecA* (and other genes)-negative isolates were identified as *Staphylococcus sciuri* (Oxa MIC = 1), and they were only weakly positive in the latex agglutination test. Possible explanations for this unexpected result may be associated with: (a) presence of a PBP2a-like protein on this isolate causing a nonspecific weak cross-reaction with the antibody used in the assay [1,14], (b) the *mecA* primers used in the study might have targeted a portion of the gene that was altered but still allowed PBP2a production [2], and (c) a false positive reaction in the PBP2a test may have occurred since the isolate was weakly positive after only 48 h incubation [65,66].

The majority of isolates (*n* = 31) still remained PBP2a-negative after repeated testing with the prolonged incubation period, of which 20 were also negative for *mecA* and other *mec* genes via PCR (Table 4), suggesting an alternative source of resistance. It has been proposed that β-lactamase hyperexpression, mutations in some PBP genes and/or their promoters, and multidrug resistance pumps such as the one encoded by the *qacC* could be responsible for oxacillin resistance in *mec*-genes-negative MRS isolates [21,31,67]. These isolates may be further studied to determine their resistance mechanism(s) in the future. The isolates negative for PBP2a but positive for *mecA* PCR (*n* = 11) suggest that either the PBP produced was so substantially altered that its binding was prevented by the antibody, or the *mecA* gene was silent, and thus PBP2a was not produced [2,14]. It is also quite possible that these isolates represent true false-negative results by the PBP2a test. In the future, it would be interesting to see whether isolates of this kind yield different results when tested with a different commercial PBP2a detection kit or following a further induction of PBP2a production by cefoxitin [20]. Lastly, as can be seen in Table 4, *mecA*-positive isolates had noticeably higher Oxa MICs compared with *mec*-gene-negative isolates, with a few exceptions, regardless of whether they were PBP2a-positive or not.

### 2.5. Genetic Diversity in MRSP

To determine the genetic diversity among a subset of methicillin-resistant *S. pseudintermedius* (MRSP), a total of 106 canine and feline isolates were randomly selected for genotyping using PFGE and MLST. Using a SmaI restriction enzyme, PFGE yielded a total of 87 unique macrorestriction patterns and 84 main PFGE profiles at the 90% similarity cutoff level, indicating an overall high level of genetic diversity among the tested MRSP population (Figure 3). Of the isolates that were of an indistinguishable genotype (*n* = 13 genotypes) or of a cluster (*n* = 3 clusters with ≥90 similarity level), about half of them (from five genotypes and three clusters) appeared to be epidemiologically unrelated (Table 5). On the other hand, isolates within seven of the indistinguishable patterns were recovered either from the same animals at the same or different time points (P3, P4, P11, P12) or/and from different animals that were treated in the same clinic (P3, P4, P7, P10, P13) as close as within the same week and as far as two months apart (Table 5). These results suggest a potential persistence of certain MRSP genotypes and nosocomial transmission among different patients in the veterinary clinics. A recent study also suggested that companion animals, people and the environment may play important roles in the transmission and persistence of MRSP in small animal primary veterinary clinics in the U.S. [68]. Supporting this view, another study indicated that MRSP frequently spread among pets in households and veterinary clinics and that the cleaning procedures utilized at the clinics were not always effective at eliminating the MRSP strains [69]. The findings from the current study also indicate the existence of certain MRSP clones with the potential to cause infections among animals (and different animal species) with no obvious epidemiological relation by location or time (P1, P5, P6, P8, P9, C1, C2, C3; Table 5). In addition, despite the limited number of cases tested, the results also indicate that the isolation of the same MRSP genotypes from the same and/or different infection sites of the same animals at the same and/or different time points (P3, P4, P11, P12; Table 5) may be a relatively common event.

MLST was performed on randomly selected isolates to represent the genetic diversity and major clusters found across the PFGE dendrogram. A total of 20 STs (including 14 known and 6 novel STs) were determined from 29 total MRSP isolates tested. The novel STs (ST1691 through ST1696) were caused by both the detection of novel sequence variants and new allelic profiles. The majority of STs (*n* = 15) were each represented by a single isolate, whereas several STs (ST71, ST181, ST833, ST1691, and ST1692) included multiple isolates (*n* = 2 to 5), with ST181 being the most common (Figure 3). Of note, ST833 (represented by two epidemiologically unrelated isolates) is not shown in Figure 3 since these isolates could not be digested with SmaI restriction enzyme despite multiple attempts. It should be emphasized that the main purpose of the MLST was not to determine the overall genetic diversity at the ST level within our collection, but rather to provide a simple means for comparison of the major lineages (based on PFGE analysis) found in this study on a global scale with respect to their host and site/lesion of isolation. The MLST-based dendrogram (Figure 4) includes the STs found in the current study and depicts their global distribution with the use of the PubMLST database (https://pubmlst.org/spseudintermedius/, accessed on 15 January 2022). As can be seen, the vast majority of the MRSP isolates from other regions around the world were associated with canine soft tissue, upper respiratory and ear infections, as was the case in our study. Not surprisingly, the most widespread global MRSP clones (ST71 in Europe and ST68 in North America) [40] were also detected among the isolates tested in this study (Figure 4); however, their true prevalence in our collection would not be accurately ascertained based on the available data. Regardless, the MLST analysis provided valuable information for comparison with previous studies and further confirmed the PFGE results indicating the existence of a high genetic diversity within the MRSP population from the Midwestern US investigated in the current study. It is also interesting to note that several STs (ST71, 155, 181, and 551) were detected both in dogs and humans (Figure 4), which further supports the notion of MRSP as being an emerging zoonotic pathogen [70,71,72].

## 3. Materials and Methods

### 3.1. Sample Source and Bacterial Isolation and Identification

Companion animal submissions available in the ISU VDL Laboratory Information Management System (LIMS) database from 2012 to 2019 were searched to determine the overall prevalence, species distribution, and antimicrobial resistance profiles of *Staphylococcus* spp. from clinical cases during the study period. The most common sample types tested were soft tissue (including skin, pyoderma, wound, and abscess), ear, urine, upper respiratory, and eye samples. The samples were processed by conventional *Staphylococcus* culture following the SOPs in place at ISU VDL. A final identification was made using MALDI-TOF mass spectrometry following the standard protocols and procedures provided by the manufacturer (Bruker Daltonics, Billerica, MA, USA). The host origin of recovered isolates (*n* = 3482 total) included dogs (83.4% of the total isolates), cats (13.4% of isolates) and other animals, such as bats, birds, horses, primates, rabbits, rodents and reptiles (3.2% of isolates).

### 3.2. Antimicrobial Susceptibility Testing

Antimicrobial susceptibility testing (AST) in the form of broth microdilution was performed using commercially available Sensititre (Thermo Fisher Scientific, Waltham, MA, USA) plates (COMPAN2F till mid-2017 and COMPGP1F thereafter) to determine the minimum inhibitory concentration (MIC) of oxacillin (Oxa) and other antimicrobials present in the panels. As Oxa was used as a proxy for ascertaining methicillin resistance, the isolates were identified as MRS when the Oxa MIC was ≥0.5 μg/mL, based on the CLSI clinical breakpoints [59,60]. The isolates were arbitrarily divided into three categories based on their Oxa MIC values and Oxa clinical resistance breakpoints for *S. pseudintermedius*/CoNS (≥0.5 μg/mL) and *S. aureus* (≥4 μg/mL): Category I (Oxa MIC <0.5 μg/mL, susceptible); Category II (Oxa MIC ≥0.5–<4 μg/mL, resistant); and Category III (Oxa MIC ≥4 μg/mL, highly resistant). Cross-resistance patterns to β-lactams and other antibiotics for *S. pseudintermedius* and CoNS, including *S. epidermidis*, were compared among the three categories. As these two Sensititre panels had different MIC ranges for some antibiotics, not all isolates could be analyzed in those cases. In addition, a subset of MRS isolates (*n* = 236) from 2016 to 2019 were selected randomly for further AST with antibiotics of importance in human medicine, including several last-line anti-staphylococci drugs, using another Sensititre panel (CMV3AGPF, Gram-positive NARMS plate). The AST results were interpreted as susceptible or not susceptible (including intermediate and resistant isolates). The MIC breakpoints (Table 6) for susceptible, intermediate, and resistant interpretive categories were mostly based on CLSI, but the European Committee on Antimicrobial Susceptibility Testing (EUCAST) breakpoints were used as needed. Tylosin has no CLSI or EUCAST MIC breakpoints for *S. pseudintermedius*; however, the resistance breakpoint ≥32 was used based on an efficiency study against *S. pseudintermedius* in dogs and a *S. aureus* bovine mastitis study [73,74].

### 3.3. Confirmation of Methicillin Resistance

Prior to testing, isolates were cultured at 35 °C for 18–24 h on TSA plates with 5% sheep blood (Remel, Thermo Fisher Scientific). Phenotypic detection of the Oxa resistance determinant, PBP2a, was performed using a PBP2a Latex Agglutination Test (Oxoid, Basingstoke, UK) on the majority of the isolates (*n* = 937 of 971 total), which were considered Oxa-resistant based on the AST results. The MRS isolates (per AST results) that were originally identified as PBP2a-negative (*n* = 36; all were CoNS except for one isolate, which was *S. aureus*) were retested following an additional 24 h incubation of cultures on blood agar to detect potential slow- or weak-PBP2a-producing isolates [20,41]. Results were interpreted as negative, positive, or indeterminate according to the manufacturer’s instructions. The isolates that still tested PPB2a-negative isolates after the retest were further retested by the AST to reconfirm their MICs. For isolates with discrepant results by these two tests, PCR was performed for the presence of methicillin-resistant genes *mecA* [77], *mecB* [25], *mecC* [30,31], and *mecD* [26], following the primer sets and protocols described in each respective publication.

### 3.4. Molecular Typing of Isolates

Pulsed-field gel electrophoresis (PFGE) was performed to determine the overall genetic diversity among representative (*n* = 106) methicillin-resistant *S. pseudintermedius* (MRSP) isolates, selected based on host species, body site, and year of isolation. Analysis of macrorestriction fragment patterns of MRSP genomic DNA using SmaI restriction enzyme was performed following previously described methods [78,79,80] with minor modifications. Briefly, the isolates retrieved from −80 °C freezer were incubated overnight at 37 °C on Mueller–Hinton (MH) agar plates (Oxoid). A well-isolated single colony from each agar plate was transferred to MH broth (Oxoid) and incubated overnight at 37 °C. The cultures were embedded in 1.2% Seakem Gold agarose (Lonza, Rockland, ME, USA) and treated with lysostaphin (Sigma-Aldrich, St Louis, MI, USA) for an hour at 37 °C in a shaking water bath, followed by incubation for 30 min at 50 °C in a static water bath in proteinase K at a concentration of 0.1 mg/mL (Sigma Aldrich). Gel plugs were washed and digested with SmaI (Promega, Madison, WI, USA) overnight at 25 °C and then embedded in 1.2% Seakem Gold agarose gel (Lonza). DNA fragments were separated using the CHEF Mapper gel electrophoresis system (Bio-Rad, Hercules, CA, USA) in 0.5× TBE buffer (at conditions 14 °C, 6 V/cm, initial switching time from 5–15 s for 8.5 h, and final switching time from 15–60 s for 11.5 h). The gel was stained with ethidium bromide for 30 min and photographed by a ChemiDoc Gel Imaging System (Bio-Rad). PFGE patterns were analyzed by the GelCompare II v.6.5 software program (Applied Maths, Kortrijk, Belgium) using the Dice similarity coefficient and unweighted-pair group method with arithmetic averages (UPGMA) with 1% optimization and 1% position tolerance. Lambda DNA ladder (Bio-Rad) was used as the molecular size marker.

Multilocus sequence typing (MLST) was carried out on 29 representative MRSP isolates (selected based on different main PFGE profiles) using previously described primers and conditions [48]. All PCR products were purified using the QIAquick^®^ PCR purification kit (QIAGEN, Hilden, Germany) and then sequenced at the DNA Core Facility of Iowa State University using an Applied Biosystems 3730xl DNA Analyzer (Thermo Fisher Scientific). Specifically, the amplification and the sequencing of the seven housekeeping genes (*ack*, *cpn60*, *fdh*, *pta*, *purA*, *sar*, *tuf*) included in the *S. pseudintermedius* scheme of the PubMLST database (https://pubmlst.org/spseudintermedius/, accessed on 1 July 2022) was performed. An MLST dendrogram (based on allelic profiles), including the sequence types (STs) found in this study and the matching STs available in the PubMLST database as of July 2020, was generated in order to provide the overall relationship of our isolates on a global scale with their STs, Oxa phenotypes, host and isolation sites, and year and country of isolation. The dendrogram was generated using iTOL without the branch lengths (https://itol.embl.de/, accessed on 15 January 2022).

## 4. Conclusions

This study provides important insights into the phenotypic and genotypic characteristics of MRS isolated from companion animal (primarily canine and to a lesser extent feline) clinical specimens submitted to the ISU VDL during a seven-year period, 2012–2019. *S. pseudintermedius* and CoNS were found to be the main *Staphylococcus* spp. associated with skin and soft tissue and various other infection types in dogs and cats, with a substantial proportion of isolates displaying resistance to methicillin along with cross-resistance to many other antibiotics of clinical significance. Importantly, our data clearly show that the prevalence of MRS from pets was steadily increasing during the study period (2012–2019). These findings underline the importance of infection control measures and routine antimicrobial susceptibility testing for successful management of MRS infections in pets, as well as for antibiotic stewardship. The continuous surveillance of MRS in veterinary settings is critical for the assessment of the concurrent resistance threat, as well as for timely detection of emerging resistance traits and the minimization of their spread in the animal–human–environment continuum.

## Figures and Tables

**Figure 1 antibiotics-11-00609-f001:**
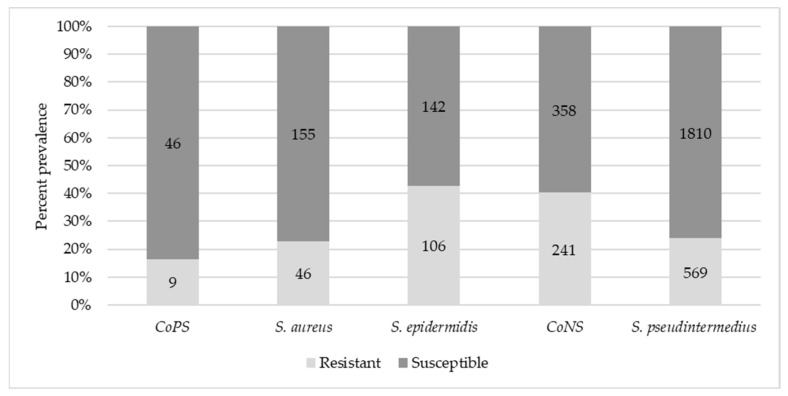
Prevalence of methicillin-resistant *Staphylococcus* spp. from companion animal clinical cases tested in this study. Total number of resistant and susceptible isolates are shown in the horizontal bars for each organism category. CoNS, coagulase-negative staphylococci excluding *S. epidermidis*; CoPS, coagulase-positive staphylococci excluding *S. aureus* and *S. pseudintermedius*.

**Figure 2 antibiotics-11-00609-f002:**
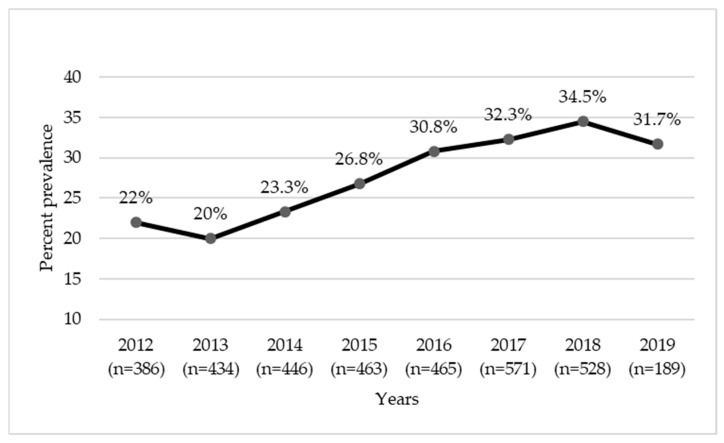
Overall methicillin-resistant *Staphylococcus* (MRS) prevalence by year (2012–2019). Total numbers of isolates (both susceptible and resistant) for each year are shown in parenthesis on the *x*-axis.

**Figure 3 antibiotics-11-00609-f003:**
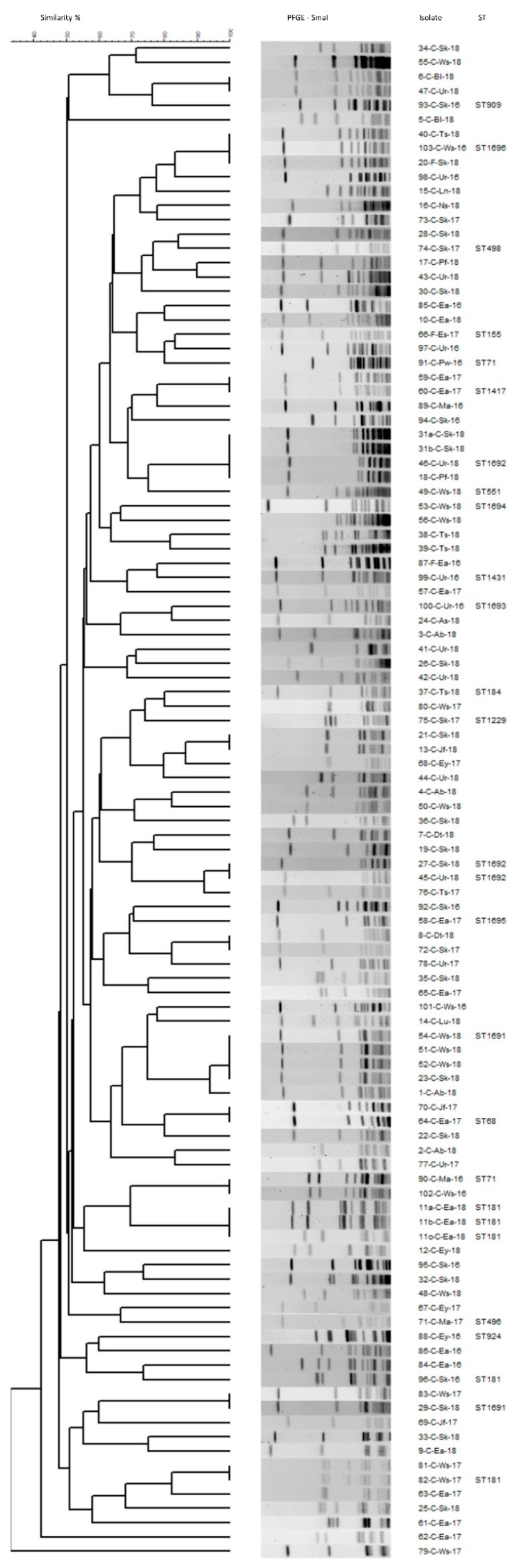
PFGE dendrogram of representative (*n* = 106) methicillin-resistant *S. pseudintermedius* (MRSP) isolates from this study. MLST-based sequence types (STs) are also shown on the far-right column when available. Isolate prefixes (from left to right): numerals, isolate numbers; C, canine; F, feline; Ab, abscess; Bl, bladder; Dt, draining tract; Ea, ear; Es, esophagus; Ey, eye swab; Jf, joint fluid; Lu, lung; Ln, lymph node; Ma, mass; Ns, nasal swab; Pf, peritoneal fluid; Pw, prostatic wash; Sk, skin; Ts, tissue; Ur, urine; Ws, wound swab; numerals, the last two digits of isolation year (e.g., 18 would mean 2018).

**Figure 4 antibiotics-11-00609-f004:**
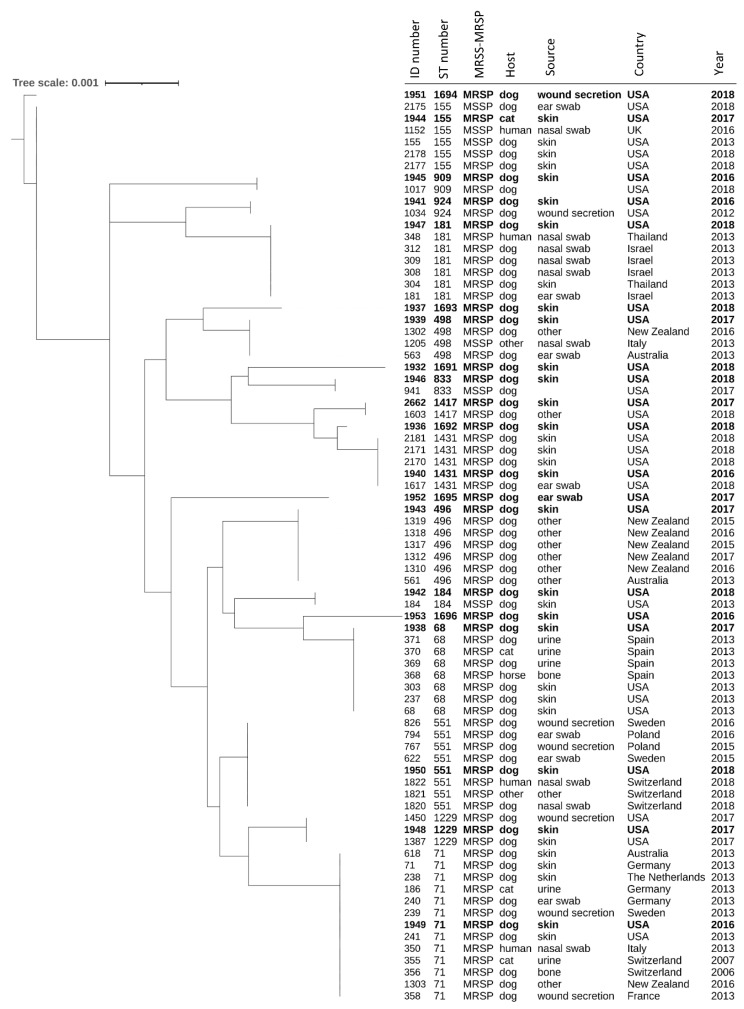
MLST-based dendrogram of the *S. pseudintermedius* STs found in the current study and their distribution on a global scale, as available in the PubMLST database. The isolates from this study are shown in boldface. The dendrogram was generated without the branch lengths and contained six datasets (PubMLST ID number, ST number, MRSP-MRSS phenotype, host, source, country, and year) using iToL (https://itol.embl.de/, accessed date 15 January 2022). In order to provide a comprehensive view, all isolates of the matching STs available in the PubMLST database (as of January 2022) were included in the dendrogram, except for ST71. For this particular ST, only 13 representative isolates (selected based on host, source, and country) from the total 54 available in the database were included for simplicity. For the STs found in this study, only one isolate (the ones submitted to the PubMLST database) per ST was included in the dendrogram. MRSP: methicillin-resistant *S. pseudintermedius*; MSSP: methicillin-susceptible *S. pseudintermedius*.

**Table 1 antibiotics-11-00609-t001:** In vitro cross-resistance patterns of *S. pseudintermedius* isolates by Oxacillin MIC.

Antibiotic Class	Antibiotic	Oxa MIC Breakpoint (µg/mL) *											
		<0.5 (S)				≥0.5–<4 (R)				≥4 (R-SA)			
		#	%NS	MIC_50_	MIC_90_	#	%NS	MIC_50_	MIC_90_	#	%NS	MIC_50_	MIC_90_
β-lactam	Ampicillin ^A^	1523	30.5	≤0.25	1	243	90.9	>1	>1	188	98.4	>1	>1
	Cefazolin ^A^	1524	0.3	≤2	≤2	243	1.6	≤2	≤2	188	57.4	4	>4
	Cefovecin ^A^	1524	2.0	≤0.5	≤0.5	243	86.0	2	>4	188	98.9	>4	>4
	Cefoxitin ^B^	1168	0.3	≤2	≤2	140	0.7	≤2	≤2	137	11.7	≤2	8
	Cefpodoxime ^A^	1524	0.5	≤2	≤2	243	30.0	≤2	8	188	94.7	>8	>8
	Ceftiofur ^C^	1168	NA	≤0.25	≤0.25	140	NA	1	4	137	NA	>4	>4
	Cephalotin ^A^	1524	0.4	≤2	≤2	243	2.1	≤2	≤2	188	45.7	≤2	>4
	Imipenem ^C^	1524	NA	≤1	≤1	243	NA	≤1	≤1	188	NA	≤1	≤1
	Penicillin ^A^	1524	51.8	0.5	8	243	93.4	>8	>8	188	98.9	>8	>8
Aminoglycoside	Amikacin ^A^	1168	0.7	≤4	≤4	141	1.4	≤4	≤4	139	5.0	≤4	8
	Gentamicin ^B^	1524	13.1	≤4	8	243	42.4	≤4	>8	188	63.8	8	>8
Fluoroquinolone	Enrofloxacin ^A^	1523	12.1	≤0.25	1	243	65.4	>2	>2	188	67.0	>2	>2
	Marbofloxacin^A^	1524	6.5	≤1	≤1	243	56.4	>2	>2	188	57.4	>2	>2
	Pradofloxacin^A^	356	8.1	≤0.25	≤0.25	103	61.2	2	>2	51	72.5	2	>2
Tetracycline	Doxycycline ^A^	768	68.8	>0.5	>0.5	195	88.2	>0.5	>0.5	132	91.7	>0.5	>0.5
	Minocycline ^C^	356	NA	≤0.5	>2	103	NA	>2	>2	51	NA	>2	>2
	Tetracycline ^A^	356	33.7	≤0.25	>1	103	78.6	>1	>1	51	80.4	>1	>1
Lincosamide	Clindamycin ^A^	1524	13.3	≤0.5	>4	243	72.8	>4	>4	188	71.8	>4	>4
Macrolide	Erythromycin ^B^	1524	14.5	≤0.5	>4	243	73.3	>4	>4	188	75.0	>4	>4
Phenicol	Chloramphenicol ^B^	1524	8.5	≤8	≤8	243	27.6	≤8	>16	188	37.8	≤8	>16
Nitrofurantion ^B^		356	0.0	≤16	≤16	103	0.0	≤16	≤16	51	2.0	≤16	≤16
Trimethoprim/Sulfamethoxazole ^B^		1524	12.1	≤2	>2	243	67.9	>2	>2	188	70.7	>2	>2

* S, susceptible; R, resistant; R-SA, resistance breakpoint by *S. aureus*; #, total number of isolates tested; NS, not susceptible (intermediate + resistant); breakpoints used: ^A^ CLSI (dogs); ^B^ CLSI (human); ^C^ NA: no breakpoint available; MIC_50_ and MIC_90_, MIC values for 50% and 90% of isolates tested, respectively.

**Table 2 antibiotics-11-00609-t002:** In vitro cross-resistance patterns of coagulase-negative staphylococci (CoNS) isolates by oxacillin MIC.

Antibiotic Class	Antibiotic	Oxa MIC Breakpoint (µg/mL) *											
		<0.5 (S)				≥0.5–<4 (R)				≥4 (R-SA)			
		#	%NS	MIC_50_	MIC_90_	#	%NS	MIC_50_	MIC_90_	#	%NS	MIC_50_	MIC_90_
β-lactam	Ampicillin ^A^	450	5.3	≤0.25	≤0.25	204	46.6	0.25	>1	103	84.5	>1	>1
	Cefazolin ^A^	450	0.2	≤2	≤2	204	4.9	≤2	≤2	101	60.4	4	>4
	Cefovecin ^A^	450	6.4	≤2	≤2	204	71.6	1	4	99	97.0	>4	>4
	Cefoxitin ^B^	353	0.9	≤2	≤2	154	22.1	4	8	78	71.8	16	>16
	Cefpodoxime ^A^	450	2.4	≤2	≤2	204	38.2	≤2	4	99	94.9	>8	>8
	Ceftiofur ^C^	353	NA	≤0.25	1	154	NA	1	2	82	NA	>4	>4
	Cephalotin ^A^	450	0.2	≤2	≤2	204	2.9	≤2	≤2	99	29.3	≤2	>4
	Imipenem ^C^	450	NA	≤1	≤1	204	NA	≤1	≤1	103	NA	≤1	2
	Penicillin ^A^	450	12.0	≤0.06	0.5	204	47.5	0.25	4	103	92.2	4	>8
Aminoglycoside	Amikacin ^A^	353	1.4	≤4	≤4	155	0.6	≤4	≤4	85	18.8	≤4	>32
	Gentamicin ^B^	450	2.2	≤4	≤4	204	8.8	≤4	≤4	103	40.0	≤4	>8
Fluoroquinolone	Enrofloxacin ^A^	450	9.3	≤0.25	0.5	204	31.9	≤0.25	>2	103	51.5	1	>2
	Marbofloxacin ^B^	450	7.3	≤1	≤1	204	30.4	≤1	>2	99	45.5	≤1	>2
	Pradofloxacin ^A^	97	3.1	≤0.25	≤0.25	50	38.0	≤0.25	2	21	28.6	≤0.25	1
Tetracycline	Doxycycline ^A^	130	49.2	≤0.12	>0.5	75	70.7	>0.5	>0.5	38	92.1	>0.5	>0.5
	Minocycline ^C^	97	NA	≤0.5	≤0.5	50	NA	≤0.5	≤0.5	21	NA	≤0.5	≤0.5
	Tetracycline ^A^	97	32.0	≤0.25	>1	50	66.0	0.5	>1	23	91.3	>1	>1
Lincosamide	Clindamycin ^A^	450	13.6	≤0.5	2	204	38.7	≤0.5	>4	99	45.4	≤0.5	>4
Macrolide	Erythromycin ^B^	450	24.7	≤0.5	>4	204	49.5	≤0.5	>4	103	49.5	≤0.5	>4
Phenicol	Chloramphenicol ^B^	450	3.1	≤8	≤8	204	2.4	≤8	≤8	103	14.6	≤8	>16
Nitrofurantion ^B^		97	3.1	≤16	≤16	50	2.0	≤16	≤16	21	14.3	≤16	64
Trimethoprim/Sulfamethoxazole ^B^		450	5.3	≤2	≤2	204	19.1	≤2	>2	103	28.1	≤2	>2

* S, susceptible; R, resistant; R-SA, resistance breakpoint by *S. aureus*; #, total number of isolates tested; NS, not susceptible (intermediate + resistant); breakpoints used: ^A^ CLSI (dogs); ^B^ CLSI (human); ^C^ NA: no breakpoint available; MIC_50_ and MIC_90_, MIC values for 50% and 90% of isolates tested, respectively.

**Table 3 antibiotics-11-00609-t003:** Distribution of MIC values of anti-Gram-positive antibiotics, including those of high importance in human medicine among MRS isolates (*n* = 236) in this study.

Antibiotic	No. of Isolates with MIC (µg/mL) ^a^	
0.03	0.06	0.12	0.25	0.5	1	2	4	8	16	32	64	≥128	MIC_50_	MIC_90_	%NS
TGC	0	60	85	71	19	 1								0.12	0.5	0.4
DAP				196	22	13	 3	0	0	0	2			0.25	0.5	2.1
LZD					14	155	54	13	 0					1	2	0.0
VAN				5	66	140	25	 0	0	0	0			1	2	0.0
QUI/DAL					205	22	 9	0	0	0	0			0.5	1	3.8
PEN				38	 10	13	10	17	10	33	105			≥16	≥16	83.9
CHL							0	35	146	 25	2	28		8	≥32	23.3
NIT							1	8	103	117	5	 2		8	16	0.8
EYR				52	35	 1	0	1	4	143				≥8	≥8	63.1
CIP			11	59	31	 20	4	3	108					1	≥4	57.2
TYL				0	36	64	22	3	2	0	 109			1	≥32	46.2
TET						 101	5	7	0	1	9	113		32	>32	100

^a^ Antibiotic concentrations included in the test panel are displayed in white areas. Thick black lines indicate breakpoints (should be read as ≥ to the corresponding concentration) for not susceptible (NS; intermediate + resistant) isolates for each antimicrobial. MICs in the gray-shaded areas should be read as ≥ to the corresponding concentration. TGC, Tigecycline; DAP, Daptomycin; LZD, Linezolid; VAN, Vancomycin; QUI/DAL, Quinupristin/dalfopristin; PEN, Penicillin; CHL, Chloramphenicol; NIT, Nitrofurantoin; EYR, Erythromycin; CIP, Ciprofloxacin; TYL, Tylosin; TET, Tetracycline.

**Table 4 antibiotics-11-00609-t004:** Detection of PBP2a protein (by a latex agglutination test) after an additional 24 h incubation of cultures and *mec* genes (by PCR) in MRS isolates (*n* = 36) that were initially tested as PBP2a negative following the standard (24 h) incubation period.

Group///Isolate	Oxa MIC	PBP2a	*mecA*	*mecB*	*mecC*	*mecD*
**PBP2a-; *mec*-** CoNS	0.5	−	−	−	−	−
*S. equorum*	0.5	−	−	−	−	−
*S. nepalensis*	0.5	−	−	−	−	−
*S. pasteuri*	0.5	−	−	−	−	−
*S. vitulinus*	0.5	−	−	−	−	−
*S. warneri*	0.5	−	−	−	−	−
*S. warneri*	0.5	−	−	−	−	−
*S. xylosus*	0.5	−	−	−	−	−
*S. xylosus*	0.5	−	−	−	−	−
*S. xylosus*	0.5	−	−	−	−	−
*S. pettenkoferi*	1	−	−	−	−	−
*S. sciuri*	1	−	−	−	−	−
*S. sciuri*	1	−	−	−	−	−
*S. sciuri*	1	−	−	−	−	−
*S. sciuri*	1	−	−	−	−	−
*S. sciuri*	1	−	−	−	−	−
*S. sciuri*	1	−	−	−	−	−
*S. sciuri*	2	−	−	−	−	−
*S. sciuri*	2	−	−	−	−	−
*S. aureus*	4	−	−	−	−	−
**PBP2a-; *mec*+** *S. haemolyticus*	0.5	−	+	NT	NT	NT
*S. haemolyticus*	>4	−	+	NT	NT	NT
*S. epidermidis*	1	−	+	NT	NT	NT
*S. epidermidis*	2	−	+	NT	NT	NT
*S. epidermidis*	2	−	+	NT	NT	NT
*S. epidermidis*	2	−	+	NT	NT	NT
*S. epidermidis*	4	−	+	NT	NT	NT
*S. epidermidis*	>4	−	+	NT	NT	NT
*S. epidermidis*	>4	−	+	NT	NT	NT
*S. epidermidis*	>4	−	+	NT	NT	NT
*S. warneri*	>4	−	+	NT	NT	NT
**PBP2a+; *mec*+** *S. simulans*	0.5	+	+	NT	NT	NT
*S. epidermidis*	4	+	+	NT	NT	NT
*S. hominis*	>4	+	+	NT	NT	NT
*S. sciuri*	>4	+	+	NT	NT	NT
**PBP2a+; *mec*-** *S. sciuri*	1	+ *	−	−	−	−

+, positive; −, negative; NT, not tested. Coagulase-negative *Staphylococcus* spp. (definitive species ID was not determined). * Weak-positive (much weaker than the positive control, but noticeably stronger than the negative control).

**Table 5 antibiotics-11-00609-t005:** Description of methicillin-resistant *S. pseudintermedius* (MRSP) isolates within the same PFGE patterns and/or main clusters *.

Indistinguishable Pattern (P)	Same Cluster (C) (≥90% Similarity)	ST	Year of Isolation	Remarks
**P1**				
40-C-Ts-18		ND	2018	Isolates in this pattern were epidemiologically unrelated
103-C-Ws-16		1696	2016	
20-F-Sk-18		ND	2018	
	**C1**			
	17-C-Pf-18	ND	2018	Epidemiologically unrelated
	43-C-Ur-18	ND	2018	
**P2**				
59-C-Ea-17		ND	2017	Related by location of owners (same town), rDVM (same), time of submissions (a week apart)
60-C-Ea-17		1417	2017	
**P3**				
31a-C-Sk-18		ND	2018	The first two are from the same animal (different sites at the same time); last two are from the same animal/different sites at the same time; first two and last two are unrelated, but treated in the same hospital two months apart
31b-C-Sk-18		ND	2018	
46-C-Ur-18		1692	2018	
18-C-Pf-18		ND	2018	
**P4**				
54-C-Ws-18		1691	2018	The first two are from the same animal/site (two months apart); last two unrelated to each other and the first two but the last three were treated in the same hospital within 2 weeks
51-C-Ws-18		ND	2018	
52-C-Ws-18		ND	2018	
23-C-Sk-18		ND	2018	
	**C2**			
	1-C-Ab-18 This isolate forms a cluster with P4	ND	2018	Epidemiologically unrelated to isolates in P4
**P5**				
70-C-Jf-17		ND		Epidemiologically unrelated
64-C-Ea-17		68		
**P6**				
27-C-Sk-18		1692	2018	Epidemiologically unrelated
45-C-Ur-18		1692	2018	
	**C3**			
	76-C-Ts-17 This isolate forms a cluster P6	ND	2017	Epidemiologically unrelated to isolates in P6
**P7**				
21-C-Sk-18		ND	2018	Treated in the same hospital within the same week
13-C-Jf-18		ND	2018	
**P8**				
8-C-Dt-18		ND	2018	Epidemiologically unrelated
72-C-Sk-17		ND	2017	
**P9**				
83-C-Ws-17		ND	2017	Epidemiologically unrelated
29-C-Sk-18		1691	2018	
**P10**				
81-C-Ws-17		ND	2017	Treated in the same hospital within the same week
82-C-WS-17		181	2017	
**P11**				
6-C-Bl-18		ND	2018	From the same animal, a few days apart, bladder stone and urine.
47-C-Ur-18		ND	2018	
**P12**				
11a-C-Ea-18		181	2018	Same animal from different ears at the same time
11b-C-Ea-18		181	2018	
11c-C-Ea-18		181	2018	
**P13**				
90-C-Ma-16		71	2016	Treated in the same hospital within the same month

* ID of isolates (starting with numerals) within each specific pattern (P1–P12) and cluster (C1–C3) are listed under the respective patterns/clusters. ND: not determined.

**Table 6 antibiotics-11-00609-t006:** The MIC interpretive breakpoints for the antibiotics used for the *Staphylococcus* spp. tested in this study.

Antibiotics	Breakpoints			
	S	I	R	Reference
Amikacin	≤4		≥16	[59]
Ampicillin	≤0.25		≥0.5	[59]
Cefazolin	≤2	4	≥8	[59]
Cefoxitin	≤4		≥8	[59]
Cefovecin	≤0.5	1	≥2	[59]
Cefpodoxime	≤2	4	≥8	[59]
Cephalothin	≤2	4	≥8	[59]
Chloramphenicol	≤8	16	≥32	[59]
Ciprofloxacin	≤0.5	1–2	≥4	[59,75]
Clindamycin	≤0.5	1–2	≥4	[59]
Daptomycin	≤1			[60]
Doxycycline	≤0.12	0.25	≥0.5	[59]
Enrofloxacin	≤0.5	2	≥4	[59,75]
Erythromycin	≤0.5	1–4	≥8	[60]
Gentamicin	≤4	8	≥16	[59]
Kanamycin			≥64	[59]
Lincomycin	≤0.5	1–2	≥4	[59]
Linezolid	≤4		≥8	[60]
Marbofloxacin	≤1	2	≥4	[59]
Nitrofurantoin	≤32	64	≥128	[60]
Penicillin	≤0.25		≥0.5	[59]
Pradofloxacin	≤0.25	0.5–1	≥2	[59]
Quinupristin/Dalfopristin	≤1	2	≥4	[60]
Tetracycline	≤0.25	0.5	≥1	[59]
Tigecycline	≤0.5		≥1	[75,76]
Trimethoprim/Sulphamethoxazole	≤2/38		≥4/76	[59]
Tylosin			≥32	[73,74]
Vancomycin	≤2 (≤4)	4–8 (8–16)	≥16 (≥32)	[59]

## Data Availability

Data are available upon request.

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
