# Peer review of "Prevalence, Mechanism, Genetic Diversity, and Cross-Resistance Patterns of Methicillin-Resistant *Staphylococcus* Isolated from Companion Animal Clinical Samples Submitted to a Veterinary Diagnostic Laboratory in the Midwestern United States"

_antibiotics, 2022, doi:10.3390/antibiotics11050609_

Round 1
Reviewer 1 Report
Authors have carried out an interesting study about different aspects of methicilllin-resistant staphylococcus isolates from companion animals from a veterinary diagnostic laboratory in the United States.
Referee acknowledge the amount of work carried out in the laboratory during seven years but it is necessary to clarify critical aspects of the paper and modify the paper to get more sound information about the research topic. I suggest major revision . Details can be found below point by point:
Introduction
It must be clearly specified the interest in genotyping/ applying molecular techniques for these strains. After reading carefully the paper two main points arise:
1.- To carry out an epidemiological link between isolates taking into account also the phenotyping profile and the molecular/genotype profile. This point is extremely relevant. In my opinion, authors must focused on this point.
2.- Try to link the different genotyping/molecular techniques with the identification of the mecA gene in the strains and the agreement between them. It could be very useful for making recommendations for interested laboratories.
Material and methods
Authors have clearly explained the sample source and antimicrobial susceptibility testing but the selection of the strains for molecular typing is not clear at all. It seems quite arbitrary and without a clear goal. My suggestion is to select the strains according to the two former points addressed by me in the introduction.
Other critical points are the different molecular/genotyping techniques applied. It makes sense to show data if different techniques are applied in the same strains to make sound comparison between them. In my opinion, it must be selected data from strains with different techniques applied.
Results and discussion
It is quite difficult to follow results and discussion in the same section. It must split this section in results for one side and discussion for the other. It could be clearer for the comprehension of the paper.
Author Response
Response to Reviewer 1 Comments
Point 1: Authors have carried out an interesting study about different aspects of methicilllin-resistant staphylococcus isolates from companion animals from a veterinary diagnostic laboratory in the United States.
Referee acknowledge the amount of work carried out in the laboratory during seven years but it is necessary to clarify critical aspects of the paper and modify the paper to get more sound information about the research topic. I suggest major revision. Details can be found below point by point:
Response 1: We sincerely appreciate the highly constructive comments and suggestions of the Reviewer. We believe that the revised manuscript has been improved significantly, and hope that it would meet the expectation of the Reviewer successfully. Our point-by-point responses are as follows.
Introduction
Point 2: It must be clearly specified the interest in genotyping/ applying molecular techniques for these strains. After reading carefully the paper two main points arise:
To carry out an epidemiological link between isolates taking into account also the phenotyping profile and the molecular/genotype profile. This point is extremely relevant. In my opinion, authors must focused on this point.
Response 2: We totally agree with the comment that both the phenotypic profiling (e.g., antimicrobial resistance and virulence) and the genotypic profiling (e.g., MLST, SCCmec types) are important tools to draw meaningful conclusion on the epidemiology of bacterial pathogens, including methicillin-resistant Staphylococcus (MRS) in companion animals. We have provided a brief description (along with several specific examples) of this subject in the revised manuscript (lines 93-109). It should be emphasized that it’s not the main goal of our study to provide an in-depth description of these associations and we believe that the interested readers could further refer to the related literature on the subject.
Point 3: Try to link the different genotyping/molecular techniques with the identification of the mecA gene in the strains and the agreement between them. It could be very useful for making recommendations for interested laboratories.
Response 3: Please see the answer above made for Comment #1.
Material and methods
Point 4: Authors have clearly explained the sample source and antimicrobial susceptibility testing but the selection of the strains for molecular typing is not clear at all. It seems quite arbitrary and without a clear goal. My suggestion is to select the strains according to the two former points addressed by me in the introduction.
Response 4: We appreciate the comment. The main purpose of the molecular typing was to determine the overall genetic diversity of the clinical methicillin-resistant Staphylococcus pseudintermedius (MRSP) isolates in our study. We first selected 106 isolates randomly for PFGE typing to represent different host species (dogs mostly and cats too), body site/lesion of isolation (e.g., ear, urine, skin, wound, abscess, etc.), and year of isolation (2016, 2017, 2018) (shown in Figure 3). We then selected 29 isolates for MLST to represent the major PFGE clusters so that we could make a comparison (of the major lineages found in our study) on a global scale with respect to host and site/lesion of isolation. We believe that we clearly pointed out these (purposes) in several places both in Material/Methods and Result/Discussion sections in the original manuscript (lines 289-314, lines 328-354, lines 425-459).
We believe that this information on the genotype diversity of the MRSP isolates from a veterinary diagnostic laboratory in the Midwestern USA provides still provides highly useful information on the molecular epidemiology of MRSP (as discussed in the original manuscript; lines 345-354). Importantly, it allowed us to make informed presumptions on the persistence, transmission, and within-animal infection dynamics of MRSP (explained in the text referring to Table 5 in the original manuscript).
Point 5: Other critical points are the different molecular/genotyping techniques applied. It makes sense to show data if different techniques are applied in the same strains to make sound comparison between them. In my opinion, it must be selected data from strains with different techniques applied.
Response 5: PFGE and MLST were applied to the same isolate collection (but not the same number of isolates) as described in the original manuscript (shown clearly in Figure 3) and as also mentioned in the answer to the first question above. These were all MRSP isolates (as stated in the manuscript). The mecA and other mec-genes PCR were done using totally different isolate collection; they were all (except for one, which was S. aureus) coagulase-negative Staphylococcus spp. (this is clearly stated in section 2.4.Mechanism of methicillin resistance in MRS and in Table 4 in the Result section in the original manuscript). We further clarified this in the Material and methods section in the revised manuscript (lines 447-459).
Results and discussion
Point 6: It is quite difficult to follow results and discussion in the same section. It must split this section in results for one side and discussion for the other. It could be clearer for the comprehension of the paper.
Response 6: We appreciate this suggestion and understand that how it could be not the most standard way of presenting the results and their significance, which may be a little bit difficult for some readers to follow. However, we also think that presenting results and discussion for each testing in tandem, at least in this manuscript, does still convey the message clearly and definitely more concisely. Also, as the Journal allows this form of format for manuscripts, we would like to keep the Results and Discussion together under one section.

Reviewer 2 Report
The manuscript „ Prevalence, mechanism, Genetic Diversity, and Cross-Resistance Patterns of Methicillin – Resistant Staphylococcus Isolated from Companion Animal Clinical Samples Submitted to a Veterinary Diagnostic Laboratory in the Midwestern United States” is a comprehensive and a well written study. Study reveals the prevalence of methicillin resistance among different staphylococcal species isolated from companion animals. Authors also discuss cross-resistance to other β-lactams as well as susceptibility of isolated strains to antibiotic for MRSA infection treatment used in human medicine. Further, authors in the study discuss the possible mechanisms of methicillin resistance as well as genetic diversity (using genotyping methods) among the strains. All the obtained results are clearly discussed and summarized in provided Tables and Figures.
There is just one observation. In section 2.1 and Figure 1, data obtained for S. aureus and S. pseudintermedius were presented separately from other CoPS. Although, it is noted in parenthesis that data for S. aureus has been excluded from data of other CoPS. What about S. pseudintermedius which is also coagulase positive bacterium?
Author Response
Response to Reviewer 2 Comments
Point 1: The manuscript „ Prevalence, mechanism, Genetic Diversity, and Cross-Resistance Patterns of Methicillin – Resistant Staphylococcus Isolated from Companion Animal Clinical Samples Submitted to a Veterinary Diagnostic Laboratory in the Midwestern United States” is a comprehensive and a well written study. Study reveals the prevalence of methicillin resistance among different staphylococcal species isolated from companion animals. Authors also discuss cross-resistance to other β-lactams as well as susceptibility of isolated strains to antibiotic for MRSA infection treatment used in human medicine. Further, authors in the study discuss the possible mechanisms of methicillin resistance as well as genetic diversity (using genotyping methods) among the strains. All the obtained results are clearly discussed and summarized in provided Tables and Figures.
Response 1: We thank the Reviewer for the comments and appreciating the value of our work.
Point 2: There is just one observation. In section 2.1 and Figure 1, data obtained for S. aureus and S. pseudintermedius were presented separately from other CoPS. Although, it is noted in parenthesis that data for S. aureus has been excluded from data of other CoPS. What about S. pseudintermedius which is also coagulase positive bacterium?
Response 2: This is a very good point, which was missed inadvertently in the original manuscript. We have clarified this point in the revised manuscript (line 123 and line 134).

Round 2
Reviewer 1 Report
Authors have addressed correctly the questions/comments addressed by the reviewer.
The paper is acceptable for publication in my opinion